# Coronary Artery Disease and Cancer: Treatment and Prognosis Regarding Gender Differences

**DOI:** 10.3390/cancers14020434

**Published:** 2022-01-16

**Authors:** Stefan A. Lange, Holger Reinecke

**Affiliations:** Department of Cardiology I—Coronary and Peripheral Vascular Disease, Heart Failure, Cardiol, University Hospital Muenster, D-48149 Muenster, Germany; holger.reinecke@ukmuenster.de

**Keywords:** cardiovascular disease, cancer, hematological malignancies, sex-specific differences, treatment, prognosis, mortality, percutaneous coronary intervention

## Abstract

**Simple Summary:**

Cardiovascular disease and cancer are the leading causes of hospitalization and mortality in high-income countries. Studies of myocardial infarction show a disadvantage for the female sex in terms of survival and development of heart failure after myocardial infarction. The extent to which this also applies to the co-occurrence of coronary heart disease and cancer was investigated and analyzed here in large registry studies. Particular attention has been paid to the four most common cancers and hematologic diseases associated with coronary artery disease requiring treatment.

**Abstract:**

Cardiovascular disease and cancer remain the leading causes of hospitalization and mortality in high-income countries. Survival after myocardial infarction has improved but there is still a difference in clinical outcome, mortality, and developing heart failure to the disadvantage of women with myocardial infarction. Most major cardiology trials and registries have excluded patients with cancer. As a result, there is only very limited information on the effects of coronary artery disease in cancer patients. In particular, the outcomes in women with cancer and coronary artery disease and its management remain empiric. We reviewed studies of over 27 million patients with coronary artery disease and cancer. Our review focused on the most important types of cancer (breast, colon, lung, prostate) and hematological malignancies with particular attention to sex-specific differences in treatment and prognosis.

## 1. Introduction

Cardiovascular disease and cancer remain the leading causes of hospitalization and mortality in high-income countries. However, since the 1980s, the incidence of acute myocardial infarction (AMI) and associated mortality has steadily declined in these countries [1]. On the other hand, the incidence of cancer, as well as the survival rate of cancer in general, increased. Thus, the number of cancer patients suffering from hemodynamically relevant coronary artery disease also increased [2,3]. This is not surprising since cancer and cardiovascular disease share some of the same risk factors such as inflammation and oxidative stress [4]. For example, the CANTOS trial of the interleukin-1ß inhibitor canakinumab demonstrated that a decrease in inflammation, as measured by a decrease in high-sensitivity C-reactive protein (hsCRP), led to a decrease in myocardial infarction, stroke, or cardiovascular death and, in particular, a decrease in the incidence of cancer [5].

The gender distribution in coronary artery disease is unfavorable to men. Three-quarters of patients with ST-elevation myocardial infarction (STEMI) and two-thirds of patients who received percutaneous coronary intervention (PCI) were men [6,7]. Survival after myocardial infarction has improved significantly over the past three decades, but there is still a difference in clinical outcomes and mortality to the disadvantage of women with AMI. Women with STEMI still have higher in-hospital mortality and are generally more likely to develop heart failure after AMI [8]. A recent study from Germany on gender differences in STEMI and NSTEMI patients between 2014 and 2017 showed disadvantages in STEMI treatment and in-hospital mortality in women, even after adjustment for age and comorbidities [9]. Specifically, women were on average approximately 12 years older and more likely to have diabetes mellitus, hypertension, and chronic kidney disease. Women with non-ST-elevation myocardial infarction (NSTEMI) were 21% less likely than men to receive PCI, and even with STEMI, the difference was 9%. In addition, women with STEMI were more likely to have a concomitant diagnosis of cancer [9].

Patients with cancer have been excluded from most large cardiology studies and registries. Therefore, there is very little information on the impact of coronary artery disease in cancer patients. In particular, outcomes in women with coronary artery disease and cancer and their treatment remain empirical. This review therefore aims to provide clinicians with an overview of how patients, particularly women, with cancer and concomitant coronary artery disease requiring treatment have been treated. We would also like to contribute to how these patients could or should be treated in the future.

Breast cancer was the most commonly diagnosed cancer in the European Union in 2020, followed by colorectal, prostate, and lung cancer [10]. Lung cancer followed by colorectal cancer and breast cancer most frequently resulted in death [10]. Therefore, our review focuses on these major cancers in particular. There are also few studies on hematologic cancers and coronary artery disease (CAD), so a subsection also addresses these. The following Medical Subject Headings (MeSH) were used to search for appropriate publications: acute coronary syndrome; myocardial infarction; non-ST-elevation myocardial infarction; ST-elevation myocardial infarction; hematologic neoplasms; leukemia; neoplasms; percutaneous coronary intervention; female. To ensure relevance of the review, included studies should not be older than 10 years at the time of publication. Publications up to April 2021 were considered, and the studies used here are summarized in Table 1: The top section compiles the studies in which 100% cancer was present but the proportion of relevant coronary artery disease was variable. The total number of patients studied here was approximately 6.8 million. The middle section lists the studies in which all patients received PCI and the proportion of cancer cases ranged from 1.27% to 23%, depending on the study. The total number of these patients was approximately 9.7 million. The bottom section compiles the studies in which there was always a myocardial infarction, with the exception of the study by Rohrmann et al. [11]. The number of patients with cancer ranged from 2.4% to 9.3%. The total number of these patients was approximately 11 million. All studies were registry studies; the majority had a retrospective design.

## 2. Cancer Patients with Acute Coronary Syndrome (ACS) in General

There is clear evidence that the risk of developing relevant coronary artery disease and AMI in cancer patients is increased [29]. In particular, the risk of this was highest in the first six months and up to one year after the initial cancer diagnosis [3,30]. This risk rapidly decreased afterwards but remained elevated 10 years and longer after the cancer diagnosis [30]. In all of the studies used here, patients with a current or previous cancer diagnosis who received PCI were up to a decade older than the general population with PCI [3,7,19,20,22,24,26,27,28,31,32].

### 2.1. In-Hospital Survival and Long-Term Outcomes after AMI in Cancer

There is clear evidence that the risk of developing relevant coronary artery disease and myocardial infarction was and is increased in cancer patients [29]. In particular, the risk of this was highest in the first six months and up to one year after the initial cancer diagnosis [3,11,30]. Thereafter, this risk declined rapidly but remained elevated for 10 years or more after cancer diagnosis [11]. In all studies used here, patients with a current or prior cancer diagnosis who received PCI were up to a decade older than the general population with PCI [3,7,19,20,22,24,26,27,28,31,32].

The mortality rate and risk of major bleeding were higher in AMI patients with a cancer diagnosis up to five years earlier than in patients without a cancer diagnosis [3]. Cancer diagnosis before AMI resulted in higher 30-day mortality, higher 1-year mortality, and worse overall survival [26]. In the majority of patients, mortality was mainly due to cancer rather than cardiac death [24,27]. Landes et al. also showed higher all-cause mortality in cancer patients after PCI. However, in this retrospective matched-pair study, the cause of death in cancer patients and noncancer patients after PCI was predominantly cardiac. Nevertheless, there was no increase in the incidence of nonfatal myocardial infarction in cancer patients after PCI [19]. AMI patients with current, active cancer had the highest risk of metastasis (20.7%) compared with AMI patients with a previous cancer diagnosis [28]. However, cancer itself does not appear to be associated with poor long-term cardiovascular outcomes in patients undergoing PCI [31]. In the presence of metastases, prostate, breast, colon, and lung cancers were independently associated with mortality, PCI complications, and major bleeding [7,28]. On the other hand, patients with metastatic cancer also had an increased risk of all-cause mortality and long-term noncardiac death compared with patients with limited cancer [3,18]. A retrospective analysis of nearly 50,000 US patients with ACS and metastatic cancer from the first decade of this century showed a beneficial effect of PCI in these cancer patients on in-hospital mortality for STEMI, but not consistently for NSTEMI [12]. This study also showed that the impact of stent implantation on in-hospital mortality decreased over time and has not been significant since 2008 [12]. Since metastatic prostate cancer was excluded from this analysis, patients who received PCI again were more likely to be younger-aged men [12]. Patients with the four most common cancers (prostate, breast, colon, and lung) who had an AMI had a higher in-hospital mortality rate without interventional treatment [28]. Patients with current or previous cancer diagnosis and AMI were less likely to be hospitalized compared with cancer-free AMI patients (current cancer, previous cancer compared with no cancer: angiography, 44.4%, 59.8% compared with 65.2%; PCI, 27.1%, 37.6% compared with 65%; and coronary artery bypass grafting (CABG) 4.9%, 7.5% compared with 9.1%) [28]. Meanwhile, the use of angiography for AMI in cancer patients, for example in Sweden, increased from approximately 33% in 2001 and 2002 to 84% in 2013 and 2014 [3].

In a small retrospective cohort, the highest in-hospital mortality after STEMI was measured in those diagnosed with cancer within 6 months previously [18]. A much larger study by Bharadwaj et al. showed a comparable association between in-hospital mortality and adverse events in patients concomitantly diagnosed with cancer, depending on the timing of cancer diagnosis. Mortality was almost twice as high in patients with current cancer as in patients without cancer. However, this was not true in patients with a history of cancer [28]. It should be noted, however, that adjusted 1-year all-cause mortality remained higher in cancer patients with AMI [26].

### 2.2. Comorbidities and Adverse Events

Complications such as serious adverse cardiac and cerebrovascular events (MACCE) and stroke were common in patients with untreated cancer and AMI. Bleeding was also more common in cancer patients, but particularly common in existing cancer and metastases [3,28]. The overall rate for myocardial infarction was inconclusive [3,26]. Stroke rates were generally similar in patients with and without cancer [26], whereas hemorrhagic strokes were slightly more common in AMI patients with a history of cancer [3].

Cancer patients with AMI also had an increased risk of being hospitalized for heart failure (HF), especially those with recently diagnosed cancer and in an advanced stage of disease compared with AMI patients without cancer [3]. In an analysis of 12 million admissions for heart failure, approximately 7% had concurrent diagnoses of lung, breast, colorectal, or prostate cancer. However, the adjusted analysis did not reveal higher in-hospital mortality in HF patients with concomitant lung, colorectal, prostate, and breast cancer compared with patients without cancer [33].

### 2.3. Gender Aspects

Women suffer AMI less frequently than men [8]. This is also one of the reasons why AMI in women is often overlooked or misjudged by physicians, which in turn leads to a longer myocardial ischemia time in women until coronary artery recanalization by PCI [34]. In addition, women were less comprehensively treated with evidence-based therapies after myocardial infarction [35]. They had poorer survival, especially if they suffered ST-elevation myocardial infarction [35]. However, it has also been shown that concomitant diseases such as diabetes and renal insufficiency were more likely to be responsible for the worse outcome after infarction compared with men than gender per se [17]. The question now arises to what extent these findings can be applied to female cancer patients with CHD. The sex ratio in cancer patients with AMI has been inconsistent in various studies [3,19,24,26,27]. In one study that also examined the difference between history of cancer and acute cancer, the proportion of females was higher in the group with history of cancer than in the group with current cancer (43% versus 35%). In comparison, the rate of female AMI patients without cancer was 39.7% [28]. However, in studies that did not distinguish between prior and current cancer, there was no consistent picture [11,16,18,21,26,32]. Patients with prior cancer were more likely to have type 2 myocardial infarction, especially those with reinfarction [3]. It is not uncommon for women with ACS and cancer to have non-obstructive coronary artery myocardial infarction (MINOCA) or Tako-Tsubo syndrome [36]. There were also gender differences in survival rates with a disadvantage for women. In a retrospective cross-sectional study from 2012 and 2014 of more than 1.13 million patients, 1.27% of whom had a co-diagnosis of cancer, women had 1.3 times higher post-PCI in-hospital all-cause mortality than men [22]. This is consistent with a meta-analysis by Pancholy et al., who found a 1.5-fold higher risk of in-hospital mortality in women with STEMI [37]. An overview of general gender differences in CHD is shown graphically in Figure 1.

## 3. Impact of Distinct Cancer Types

### 3.1. Prostate Cancer

In 2020, prostate cancer had the highest incidence rate in the European Union and ranked fifth in cancer-related mortality at 5.5% [10]. The risk of coronary heart disease was shown to be significantly increased in prostate cancer patients in the first six months after cancer diagnosis (standardized incidence ratio, SIR is 1.41; 95% CI 1.33–1.49) and remained consistently elevated for up to 10 years (SIR was 1.13; 95% CI 1.1–1.16) [3,28].

*Angiography and PCI.* PCI for current prostate cancer was not associated with increased in-hospital mortality compared with non-cancer patients but was associated with an increased risk of bleeding (odds ratio, OR 1.41, 95% CI 1.2–1.65 for current and 1.20, 95% CI 1.10–1.31 for historical) [3,7]. In addition, AMI patients with prostate cancer were less likely to receive coronary angiography (47.5% vs. 65.2%) and less likely to receive PCI (29.3% vs. 43.9%) compared with AMI patients without cancer [28]. However, despite the risk of bleeding during prolonged dual antiplatelet therapy (DAPT), drug-eluting stents (DES) were implanted more frequently than bare metal stents (BMS) (current: DES 63.3% vs. BMS 31.5%, historical: DES 73.0% vs. BMS 23.0%). The indication for coronary angiography and PCI was more stringent in prostate cancer patients with a current diagnosis than in cancer patients with a historical diagnosis, as evidenced by the fact that STEMI was a more frequent indication in current prostate cancer than in patients with historical cancer (19.8% and 18.4%, respectively) [28].

*Co-morbidities, adverse events and survival.* Bleeding occurred more frequently in prostate cancer patients with AMI than in patients without cancer both during the inpatient stay (+44%) and within 90 days of discharge [28]. However, the incidence of bleeding in prostate cancer patients leading to inpatient readmission was significantly increased within 90 days after PCI only when there was concomitant metastatic spread [38].

Therefore, it is good to know that metastasis is expected in 9.2% of current prostate cancer patients and only in 1.2% of historical prostate cancer patients [11]. Readmission within 90 days after index PCI due to another AMI was significantly increased only in metastatic prostate cancer [38]. In addition, anemia as a relevant comorbidity during PCI occurred more frequently in patients with acute prostate cancer than in patients with a historical prostate cancer diagnosis (13.9% vs. 10.5%) [28].

Heart failure was more common in prostate cancer patients with AMI compared with patients without cancer [3]. Among these, most patients had a current and, less frequently, a historical diagnosis of prostate cancer [28]. All-cause mortality was higher in prostate cancer with myocardial infarction than in infarction patients without cancer [3]. The odds ratios for mortality and MACCE were 1.19 and 1.17, respectively, for AMI and concurrent prostate cancer [28]. Looking at mortality in heart failure, a study of about 12 million patients with heart failure between 2003 and 2014 showed that overall in-hospital mortality was 3.3% while mortality in concurrent prostate cancer was slightly higher at 3.5% [33].

*Summary*. To assess CAD in the presence of concurrent prostate cancer, a fundamental distinction must be made between current and previous prostate cancer. CHD patients with acute prostate cancer have a significantly worse prognosis compared to patients with a history of prostate cancer or to CAD patients without cancer. Special attention should be paid to the high-risk group of patients with metastatic prostate cancer, who also had more frequent recurrences of myocardial infarction and bleeding during the course of the disease.

### 3.2. Breast Cancer

Of all cancers, breast cancer had the highest cancer incidence rate among women living in the European Union in 2020 and ranks first in cancer-related mortality among women [10]. Patients with breast cancer had an increased risk of coronary heart disease exclusively in the first six months after cancer diagnosis (standardized incidence ratio, SIR 1.27; 95% CI 1.14–1.41) [28]. Fortunately, the risk of myocardial infarction was not increased in breast cancer patients [3]. Although breast cancer also occurs in men (0.6–1.0%) [39], most patients with current or historical breast cancer who received PCI were almost exclusively women in the trials (99.8%, both) [7,28]. This may also be due to the overall poor clinical outcome of breast cancer in men [39]

*Angiography and PCI.* Similar to prostate cancer patients, breast cancer patients with AMI were less likely to receive coronary angiography (47.0% vs. 65.2%), PCI (27.4% vs. 43.9%), and CABG (4.2% vs. 9.1%) compared with AMI patients without cancer [28]. Among patients hospitalized for STEMI in the United States between 2001 and 2011, the rate of PCI was only 30.8% in patients with active breast cancer. Unfortunately, breast cancer patients with STEMI who did not receive PCI had a 3.5-fold higher hospital mortality than breast cancer patients with STEMI and PCI [25]. It should be emphasized that a current breast cancer treated with PCI was not significantly associated with increased in-hospital mortality or complications [7].

After PCI, the rate of readmissions within 90-days with AMI was significantly increased by 7.5% in patients with active breast cancer compared with non-cancer patients, whereas readmissions due to bleeding did not differ [23]. However, when adjusted readmission rates are considered, it is evident that breast cancer with or without metastases had no effect on readmission due to AMI or bleeding, in contrast to prostate cancer [23]. The reason for PCI in current vs. historical breast cancer was STEMI during hospitalization in 22.6% vs. 19.1%, respectively [7]. DES were implanted more frequently than BMS in these patients (in current: 57.1% DES vs. 36.1% BMS, in historical: 73.1% DES vs. 22.4% BMS) [7].

*Co-morbidities, adverse events and survival.* Congestive heart failure occurred in 2.4% of current breast cancer and 0.9% of historical breast cancer [7]. AMI survivors with breast cancer were more likely to develop congestive heart failure during follow-up than AMI survivors without cancer (hazard ratio, HR 1.33; 95% CI, 1.11–1.58; *p* = 0.0016) [26]. In contrast, Velders et al. did not show an increased HR for hospitalization for heart failure in breast cancer patients with AMI compared with patients with AMI without cancer [3]. Anemia as a relevant comorbidity after PCI occurred in 17.9% of patients with an acute breast cancer diagnosis and in 14.0% with a historical breast cancer diagnosis [7]. Among patients with PCI and breast cancer, metastases were detected in 15.1% with current breast cancer and 1.4% with historical breast cancer [7]. Patients with metastatic breast cancer and concurrent ACS were more likely to undergo PCI than was the case with other metastatic cancers (e.g., metastatic lung and pancreatic cancer) [12]. However, it should also be noted that the overall mortality of AMI patients with breast cancer was higher than in patients without cancer [3]. The OR was 1.31 for in-hospital mortality in breast cancer patients with AMI and 1.23 for MACCE. The incidence of bleeding in breast cancer patients with AMI varied from study to study. While Bharadwaj et al. recorded an OR for the frequency of bleeding of 1.29, Velders et al. no significantly increased HR was found for bleeding of any kind [3,28].

*Summary.* In general, the indication for PCI for AMI in patients with breast cancer is hardly limited, especially when acute infarction is present. Even metastasis does not significantly increase the risk of a bleeding complication.

### 3.3. Colorectal Cancer

The incidence of colorectal cancer ranked second in the European Union (EU) in 2020, regardless of gender and age. The incidence was about 63% higher in men than in women. In cancer-related mortality, colorectal cancer ranked second regardless of gender [10].

The risk of coronary heart disease was significantly increased in colorectal cancer patients in the first six months after cancer diagnosis (SIR 1.82; 95% CI 1.66–1.99), but remained consistently elevated beyond that for up to 10 years (SIR 1.06; 95% CI 1.02–1.11 [30]. Nevertheless, the risk of myocardial infarction was not increased in colorectal cancer patients [3].

*Angiography and PCI*. Compared with AMI patients without cancer, patients with colorectal cancer were less likely to receive coronary angiography (44.7% vs. 65.2%), less likely to receive PCI (27.6% vs. 43.9%), and less likely to receive CABG (5.1% vs. 9.1%) [28]. Even among patients with STEMI hospitalized in the United States between 2001 and 2011, PCI was performed in only 17.3% of cases [25]. This is unfortunate because in-hospital mortality was more than three times higher in STEMI patients with colorectal cancer who did not receive PCI [25]. Of note, patients with colorectal cancer who received PCI did not have increased in-hospital mortality (OR 1.39, 95% CI 0.99–1.95), but did have increased rates of in-hospital complications (OR 2.17, 95% CI 1.9–2.48), bleeding (OR 3.65, 95% CI 3.07–4.35), and cardiac complications (OR 1.45, 95% CI 1.16–1.81) compared with patients without colorectal cancer [7]. Patients with colon cancer who received PCI were mostly male (64.2%, both) [7,28]. PCI was performed more frequently in patients with ACS and metastatic colon cancer compared to cancers with a significantly poorer prognosis (e.g., metastatic lung and pancreatic cancer) [12]. In current colorectal cancer patients, DES were implanted less frequently than BMS (38.4% DES vs. 46.8% BMS). In contrast, DES were preferred in patients with a history of colorectal cancer (68.9% DES vs. 26.8% BMS). The indication for PCI in patients with current or historical colorectal cancer diagnosis was STEMI in 26.0 and 18.1%, respectively [7].

*Co-morbidities, adverse events and survival.* The 90-day readmission rate after PCI due to recurrent myocardial infarction (10.8%) or hemorrhage (4.2%) were significantly increased in patients with both limited and metastatic colorectal cancer compared with patients without cancer [23]. However, it should be noted that in this study, the risk of reinfarction was more than twice that of hemorrhage [23]. MACCE were more common in patients with colon cancer and AMI than in patients without cancer (OR 1.49). However, a particularly common adverse event in colorectal cancer with AMI was bleeding (OR 2.82) [28]. In contrast, a registry from Sweden showed no significantly increased hazard ratio for bleeding in general in patients with AMI and colorectal cancer [3]. Anemia under PCI as a relevant comorbidity occurred in 34.1% of patients with an acute colorectal cancer diagnosis. In contrast, anemia under PCI occurred in only 14.6% of patients with a previous diagnosis of colorectal cancer [7]. Congestive heart failure occurred in 8.8% of patients with a current colorectal cancer diagnosis and 0.9% with a historical colorectal cancer diagnosis [7]. In contrast, a study from Sweden showed no increased hazard ratio for hospitalization for heart failure in AMI patients with colorectal cancer [3]. It should be emphasized that metastases were found in 24.1% of patients with current colorectal cancer and 2.1% of patients with previous colorectal cancer who received PCI [7].

In-hospital mortality was particularly high in patients with colorectal cancer and AMI compared with patients without cancer (11.6%) [28].

*Summary*. Patients with colorectal carcinoma who underwent PCI had a significantly higher complication rate, with bleeding playing a particular role. However, the risk for reinfarction was also significantly higher than that for bleeding.

### 3.4. Lung Cancer

Lung cancer ranked fourth in incidence for all ages for both sexes in the EU in 2020, but lung cancer ranked first in cancer-related mortality regardless of sex [10].

Additionally, the risk of coronary heart disease in the first six months after diagnosis was significantly higher for lung cancer (SIR 2.56; 95% CI 2.35–2.80) [30].

*Angiography and PCI.* Current lung cancer patients with PCI had a significantly increased in-hospital mortality rate (OR 2.81, 95% CI 2.37–3.34), as well as an increased rate of in-hospital complications (OR 1.21, 95% CI 1.1–1.36) and bleeding (OR 1.79, 95% CI 1.56–2.05) compared to patients without cancer [7]. AMI patients with lung cancer were less likely to receive coronary angiography (34.8% vs. 65.2%), PCI (21.0% vs. 43.9%), and CABG (2.3% vs. 9.1%) compared with AMI patients without cancer [28]. Even in STEMI patients with lung cancer, the proportion of patients who received PCI again was only about 20% [25]. Lung cancer patients lagged behind the most common cancers in such interventions [28]. STEMI patients with lung cancer who did not receive PCI had approximately 2.7 times higher in-hospital mortality than lung cancer patients with STEMI [25]. Presumably because of the generally poor survival in metastatic lung cancer, these patients historically received PCI when NSTEMI occurred and even less frequently when STEMI occurred compared with patients without cancer [12]. Patients with current or past lung cancer who received PCI were mostly male (65.3% and 61.5%, respectively) [7,28].

In patients with current lung cancer, DES were implanted less frequently than BMS (39.3% DES vs. 49.6% BMS). In contrast, DES were preferred in patients with a previous lung cancer diagnosis (67.1% DES vs. 27.6% BMS) [7]. The reason for PCI in current vs. previous lung cancer was STEMI during hospitalization in 30.4% and 17.9%, respectively [7]. Again, this demonstrates the particularly stringent consideration of indications for coronary angiography and PCI. The 90-day readmission rate for AMI after initial PCI was strikingly high (12.1%) compared with prostate, breast, and any cancer in lung cancer.

*Co-morbidities, adverse events and survival.* AMI Patienten mit der Diagnose Lungenkrebs in der Vorgeschichte hatten eine erhöhte Krankenhausmortalität (OR 1.65, 95% CI 1.32–2.05) und eine erhöhte Rate an Blutungen (OR 1.18, 95% CI 1.01–1.38) [7]. Eine Anämie als relevante Komorbidität trat bei 21.4% der aktuellen und 12.5% der historischen Lungenkrebspatienten nach PCI auf [7]. Hingegen war die 90-Tage-Rückübernahme wegen Blutungen bei aktivem Lungenkrebs (1.5%) war vergleichbar mit anderen Krebsarten (mit Ausnahme von Dickdarmkrebs), und nach Anpassung der Quoten nicht statistisch signifikant [23]. Eine kongestive Herzinsuffizienz trat bei 5.2% bei aktuellem und 1.8% bei historischem Lungenkrebs auf [7]. Metastasen wurden bei 23.2% bei aktuellem und 2.1% bei historischem Lungenkrebs nach PCI gefunden [7]. Die häufigste Todesursache bei Lungenkrebs-”Überlebenden” war primärer Lungenkrebs, selbst 20 Jahre nach der Erstdiagnose Lungenkrebs [40]. Es besteht eine hohe In-Hospital-Mortalität und MACCE Rate bei Patienten mit Lungenkrebs und AMI (OR 2.71 bzw. 2.38). Bleeding (OR 2.06) and stroke also occurred more frequently (OR 1.91) than in patients without cancer [28].

*Summary.* Lung cancer patients have a very poor prognosis overall compared to other cancers. Complications such as severe bleeding and strokes certainly contribute to this. In contrast with other cancers, primary lung cancer is also the most common cause of death.

### 3.5. Hematological Malignancies

*Leukemia and CHD in general.* The incidence of leukemia was 14.1 per 100,000 population in the EU in 2020 [10]. The incidence was 1.62 times higher in men than in women [10]. Leukemia accounted for 3.4% of all cancer-related mortality, which is relatively low for both sexes and all age groups [10]. Patients with AMI and a current leukemia diagnosis were rarely encountered. They accounted for only 0.3% of patients with AMI [14]. Conversely, 1.4% of patients with malignant hematologic disease suffered an ACS [13]. Myocardial infarction occurred frequently in patients with hematologic cancer [3]. However, in a large Swedish registry, patients with leukemia had a marked risk of coronary heart disease in the first 6 months after diagnosis. Their standardized incidence ratio for coronary heart disease was 2.81 (95% CI 2.37–3.37) [30]. An analysis of over 21,000 patients with a primary discharge diagnosis of AMI and a concomitant leukemia diagnosis from the U.S. National Inpatient Sample between 2004 and 2014 showed that these patients were approximately 10 years older than AMI patients without leukemia (77 vs. 68 years) and predominantly male, especially in acute leukemia. Chronic lymphocytic leukemia (CLL) accounts for the largest proportion with over 70%, followed by chronic myeloid leukemia (CML) with almost 16%, acute myeloid leukemia (AML) with 10.5%, and acute lymphoblastic leukemia (ALL) with 1.7% at last [3,14].

*Multiple myeloma, non-Hodgkin’s lymphoma (NHL), and Hodgkin’s disease.* Statistically, the incidence of multiple myeloma in 2020 in the EU-10 was 7.5/100,000 for both sexes. The risk of coronary heart disease was higher in patients with multiple myeloma in the first six months after cancer diagnosis (standardized incidence rate 2.17; 95% CI 1.77–2.63) and remained consistently elevated for up to 10 years (standardized incidence rate 1.36; 95% CI 1.24–1.55) [30]. NHL had an incidence of 18.3/100,000 people in the EU in 2020. In contrast, the incidence for Hodgkin’s disease in the same region and in the same year was 2.7/100,000 people, just under one-seventh in comparison. Both NHL and Hodgkin’s disease affected men significantly more often than women (approximately 3:2) [10]. The risk of coronary artery disease was increased in NHL patients in the first six months after diagnosis (standardized incidence ratio 2.2; 95% CI 1.96–2.46), but decreased thereafter [30]. Encouragingly, the risk of coronary heart disease was not increased in Hodgkin’s disease [30].

*Angiography and PCI in hematologigical malignancies.* Coronary angiography and PCI or CABG have not been performed very frequently in patients with leukemia, especially in patients with AML. [13,14]. Those referred for coronary angiography had higher hemoglobin levels, higher platelet counts, and lower leukocyte counts [14]. In the very small retrospective study by Park et al., in which only 25% of all patients (total group, n = 73) underwent coronary angiography, one-third had intracoronary thrombus, and half of the patients had severe three-vessel or left main coronary disease. Approximately 75% of ACS patients were treated with drugs alone [13].

*Co-morbidities, adverse events and survival.* Patients with leukemia had a higher risk of MACCE and increased all-cause mortality compared with AMI patients without malignancy [3,14]. The risk of bleeding in hematologic cancer was not clear in the few studies. Mohamed et al. did not find an increased risk of bleeding in hematologic cancers compared with AMI patients without cancer [14]. In contrast, using a large Swedish database on AMI and cancer, Velders et al. demonstrated that the risk of fatal or nonfatal bleeding complications was higher in patients with leukemia compared with other cancers [3].

AML was associated with approximately three times the risk of myocardial infarction and four times the risk of death compared with patients without AML [14]. In-hospital and especially 1-year mortality were quite high in patients with hematologic diseases and myocardial infarction, although in this context 75% of in-hospital mortality and 85% of 1-year mortality, respectively, could not be attributed to a cardiac origin [13]. It has also been shown that the risk of hospitalization for heart failure was particularly high in AMI patients with hematologic cancers compared with patients without such conditions [3]. Other features of leukemias include an accumulation of thrombocytopenias, coagulopathies, and deficit anemias [14]. Thrombocytopenia occurred particularly in more than a quarter of patients with acute myeloid leukemia and AMI. Thus, it is not surprising that although most patients received a beta-blocker, less than 50% of non-interventional patients received an antiplatelet agent or anticoagulants [13].

*Summary:* Leukemia is a very heterogeneous group of diseases, representing only a small fraction of cancers. Of these, AML is associated with a particularly high risk of thrombocytopenia, coagulopathies, and anemia, and also has a very poor prognosis.

A general rejection nor a nonselective use of invasive treatments such as coronary angiography and PCI should be recommended in leukemia and ACS because of the current limited data.

A compilation for the treatment of coronary artery disease in patients with and without cancer is shown in Table 2. A summary of the prevalence of adverse events and complications associated with PCI in cancer patients is shown in Table 3.

## 4. Specific Cancer Treatment and Cardiovascular Side-Effects

Nowadays, depending on the type of cancer, modern cancer therapy consists of different groups of drugs in combination with radiotherapy. Unfortunately, most cancer therapies also have a cardiotoxic component. Tumor therapies that can induce acute coronary syndrome include alkylating agents, antimetabolites, anti-microtubule agents, antibiotics, hormonal therapies, monoclonal antibodies, and tyrosine kinase inhibitors (TKIs) [41,42,43]. Radiation therapy used for certain types of tumors in the chest can damage the vascular walls and lead to thrombosis, plaque formation, and fibrosis [44,45,46]. Thus, mediastinal fibrosis, aortic valve disease, and coronary artery disease are also consequences of such chest irradiation [38,46,47]. In addition, there are also gender differences in the absorption and distribution of cancer-specific drugs. Women have a greater volume of distribution for lipophilic drugs, whereas men have a greater volume of distribution for water-soluble drugs [48]. There are also significant sex differences in renal function. On average, men have approximately 20% higher creatinine clearance than women [49]. Last but not least, men tend to have higher CYP1A2, CYP2D6, and CYP2E1 activity, resulting in increased metabolism of the drug substrates of these enzymes [48]. In contrast, females exhibit higher CYP3A4 activity, the most abundant cytochrome isoenzyme involved in the metabolism of most drugs [48]. Moreover, in gender comparison, phase II metabolism of drugs by UDP-glucuronosyltransferase, sulfotransferases, and N-acetyltransferases is increased in men [48]).

In addition, the relationship between the tumor diseases primarily described here and the specific tumor therapy used in each case today and its influence on coronary heart disease will also be discussed under the aspect of gender differences in Section 4.1, Section 4.2, Section 4.3, Section 4.4 and Section 4.5. To avoid redundancy, tumor-specific drugs already mentioned are not listed again.

### 4.1. Prostate Cancer

Anti-hormone therapy is very commonly used in patients with prostate cancer. These include antiandrogens such as bicalbutide and gonadotropin-releasing hormones such as goserelin and degarelix. The cardiovascular side effects of this hormone therapy include progression of coronary artery disease, angina pectoris, and ACS [50,51]. Apalutamide and darolutamide, two new androgen receptor antagonists now widely used in prostate cancer therapy, also showed cardiovascular side effects in varying severity and frequencies.

This also includes high blood pressure, heart failure, cardiac arrest, coronary artery disease, angina pectoris, and acute myocardial infarction [52,53,54]. On the other hand, enzalutamide, another new androgen receptor antagonist, did not increase the risk of cardiac events, but it did increase the risk of hypertension [55]. The increase in cardiovascular events can be explained by the fact that this androgen-depriving therapy can lead to dyslipidemia, sarcopenic obesity, and insulin resistance with diabetes mellitus [54]. Abiraterone a CYP17 inhibitor in the modern prostate cancer therapy was also associated with increased risk of cardiac events, atrial tachyarrhythmias, heart failure, and the risk of hypertension [55].

### 4.2. Breast Cancer

In addition to the use of 5-fluorouracil (5-FU) as an antimetabolite and the anti-microtubule agents such as vinblastine, aromatase inhibitors such as anastrozole are also used in breast cancer. These chemotherapeutic agents can also induce angina, vasospasm, and ACS [56,57,58]. It should not go unmentioned that women have a longer lasting effect level of 5-FU compared to men and thus a higher toxicity (AUC female vs. male 22 vs. 18 mg h/L, *p* = 0.04) [59]. A meta-analysis by Khosrow-Khavar et al. showed that aromatase inhibitors are not associated with an increased risk of cardiovascular events. Even adjuvant treatment with tamoxifen resulted in a 33% reduction in the risk of cardiovascular events [60]. In contrast, the VEGF monoclonal antibody bevacizumab nearly doubles the risk of cardiac ischemia at high doses [61]. It is therefore particularly important to note that women have approximately 20% lower clearance of bivacizumab than men [48]. Another specific treatment for HER2-positive breast cancer is monoclonal antibodies such as trastuzumab, pertuzumab, and corresponding antibody conjugates. Relevant cardiotoxicity occurs in approximately 10% of patients treated with these agents. This cardiotoxicity is manifested by a decrease in LV function with or without clinical signs of heart failure [62,63]. However, an increase in myocardial infarction due to this antibody therapy is not observed [63].

### 4.3. Colon Cancer

Capecitabine, a pro-drug of 5-FU, and oxaliplatin, an alkylating agent, are part of a chemotherapy protocol for advanced colon cancer [64]. Capecitabine can induce Angina, Vasospasm and ACS [57]. Oxaliplatin can induce acute ST segment elevation without myocardial infarction [65]. However, as part of adjuvant treatment, oxaliplatin and 5-FU had no-effect on cardiac function [66].

### 4.4. Lung Cancer

A common therapy for advanced lung cancer is a combination of cisplatin, etoposide, vinorelbine, gemcitabine and taxanes [67]. With cisplatin, myocarditis, pericarditis, angina, and myocardial infarction have been observed [68]. Paclitaxel and Docetaxel were common with vasospasm, ACS and bradycardia [69]. Reduced clearance was also documented for paclitaxel in women [48]. Etoposide and bleomycin have been associated with acute myocardial infarction in young patients without coronary artery disease [70]. Cases of acute myocardial infarction have also been described after treatment with vinorelbine, a vinca alkaloid, and with gemcitabine [71,72,73,74,75].

### 4.5. Hematological Malignancies

The following chemotherapeutic agents are used for various types of lymphoma and may in turn cause angina, vasospasm, and ACS: Vinblastine, bleomycin, and rituximab [41]. Rituximab can also induce takotsubo cardiomyopathy [76].

Multiple myeloma. TKI therapy resulted in progression of coronary artery disease and ACS [77]. However, the risk of cardiovascular events is higher with second- (dasatinib, nilotinib, or bosutinib) and third- (ponatinib) generation TKIs than with imatinib, the first-generation TKI. Unfortunately, the TKIs are also associated with thrombocytopenia [78]. Non-Hodgkin’s lymphoma and Hodgkin’s disease. Antitumor antibiotic treatment with bleomycin for non-Hodgkin’s and Hodgkin’s lymphomas can cause angina, vasospasm, and ACS [56]. Cyclophosphamide, part of the CHOP treatment protocol (Cytoxan, hydroxyrubicin (Adriamycin), Oncovin (Vincristine), Prednisone (chemotherapy regimen), can cause left ventricular dysfunction, heart failure, myocarditis, pericarditis, arterial thrombosis, arrhythmias such as bradycardia, atrial fibrillation, and supraventricular tachycardia [79]. With rituximab and imatinib, decreased drug clearance in women should be noted [48].

## 5. Outcomes of Revascularization Procedures in Cancer Patients

Invasive therapies, e.g., coronary angiography, PCI, and CABG surgery, were similarly used in patients with or without cancer, but the frequency increased dramatically between 1995 and 2013. In particular, the increase factor for PCI in AMI was 11.4 without cancer comorbidity and 12.6 with cancer comorbidity [26]. Nakatsuma K. et al. were able to show a higher 5-year cumulative incidence rate of all-cause death, noncardiac death, and cardiac death after stent implantation in patients with a history of cancer, even after adjusting for confounders, in the coronary revascularization demonstrating outcome study in Kyoto registry cohort-2 between 2005 and 2007. In this study, drug-eluting stents were used in 51% of cancer patients compared with 56% DES in general [20]. The use of BMS and POBA was more common in cancer patients undergoing PCI than in cancer-free patients [19]. There was a tendency for cancer patients to have definite or probable stent thrombosis and a significantly increased risk of major bleeding, but no increase in myocardial infarction and stroke [20].

Compared with the first decade of this century, cancer patients who underwent PCI after 2011 showed improved overall survival but no significant reduction in major adverse cardiac events (MACE) [15]. The safe access route via the radial artery could be a building block for this. Transradial access should be preferred, especially in patients with potential bleeding risk such as patients with ACS and cancer. Only in cancer patients on hemodialysis, abnormal Allen tests, or post-mastectomy women must radially access on the appropriate side be avoided and alternative access routes used [80]. Therefore, it is not surprising that a significant increase in radial access for PCI in cancer patients was observed after 2011 [15]. As shown in most studies, the risk of minor and major bleeding increases in cancer patients, especially those with blood and gastrointestinal cancers who have undergone coronary therapy. Therefore, for these patients, it would be better to keep DAPT as short as possible after stent implantation, especially in patients with an existing or impending risk of thrombocytopenia, such as hematologic cancers [78]. It is reassuring to know that the recommendations for aspirin apply up to a platelet count of 10,000/mL [81]. A consensus paper on cardiac catheterization in oncologic patients summarizes in detail the measures required for thrombocytopenia. The main recommendations are summarized in Table 4.

However, these 2016 recommendations should be revised in the near future, as new stenting techniques, studies, and meta-analyses of DAPT after PCI will provide new opportunities and insights. POBA-only and the use of BMS allow the duration of DAPT to be kept short. However, there are prognostic disadvantages and risks of restenosis when using POBA or BMS compared with using DES. The solution could be an abluminal drug-eluting stent technology that can shorten DAPT to one month. The comparison between BMS and a new polymer-free drug-eluting coronary stent showed that this stent technology was superior to bare metal stents in terms of myocardial infarction and required revascularization procedures at the same risk of bleeding [82]. This study also included approximately 10% cancer patients, and this very small subgroup of 239 patients showed only a trend toward a better primary safety and efficacy endpoint [82]. A recently published study comparing polymer-free and polymer-based stent technology failed to show inferiority of the latter but older stent design for the primary safety combination endpoint of death from cardiac progression, myocardial infarction, or stent thrombosis at 1 year [83]. Thus, it can be concluded that appropriate drug-eluting stent technology is preferable to BMS in all cases, even in patients such as cancer patients with platelet counts greater than 20,000/mL. Nevertheless, the duration of antiplatelet therapy after PCI and DES implantation remains controversial. In a meta-analysis by Benenati, S. et al., short (1 or 3 months) DAPT was compared with long (12 months) DAPT in patients undergoing PCI. They concluded that very short DAPT did not increase the risk of ischemic complications but reduced the risk of bleeding [84]. A recent meta-analysis by Xu, Y. et al. comparing short (1–3 months followed by acetylsalicylic acid, ASA, or P2Y12), intermediate (6 months), long (12 months), and extended long-term (more than 12 months) DAPT after PCI with DES also showed no differences in coronary ischemic events between short, intermediate, and long-term DAPT. However, fewer ischemic adverse events occurred with prolonged DAPT [85]. The risk of bleeding was significantly lower with short-term DAPT followed by P2Y12 than with 12-month DAPT [85]. One option is to stop dual antiplatelet treatment early, e.g., three months after DES implantation when using 3rd generation DES. According to the “Twilight Study,” another option is to continue using the guanosine 5’-diphosphate inhibitor (GDP) ticagrelor and discontinue ASA instead [86]. However, cancer patients were not specifically mentioned in the study. A potential decision aid for early discontinuation of dual platelet inhibition, if appropriate, is invasive examination of the healing process of a stented coronary artery with assessment of neointima formation and possible stent malposition using optical coherence tomography (OCT). This technique provides excellent visualization of the coronary artery wall even with the stent in place [87,88]. 

An overview of the expected risk for bleeding, cardiac complications/MACCE, and in-hospital mortality during PCI and AMI in patients with prostate cancer, breast cancer, colorectal cancer, lung cancer, leukemia, and no cancer is summarized graphically in Figure 2.

For an orienting guide to the decision for or against percutaneous coronary intervention in cancer, see Figure 3.

## 6. Conclusions

Cancer and cardiovascular disease are closely related in their origins [4]. The likelihood of concomitant coronary heart disease is higher in cancer patients than in the general population without cancer [29,30].

Unfortunately, women with CHD are still disadvantaged in the prognosis and treatment of acute myocardial infarction [8]. There are few studies that have specifically examined gender differences in cancer and CHD. When they did, they did not distinguish between the different types of cancer. The proportion of women in this group of patients with cancer and CHD was higher in most studies [3,24,26], but treatment for CHD did not differ significantly between patients with and without cancer [3,19,24,26,27]. However, cancer patients with myocardial infarction were also significantly older and had more concomitant diseases [3,7,19,20,22,24,26,27,28,31,32].

Regardless of the underlying cancer, patients with concomitant infarction had a higher mortality rate [3,26]. However, in most studies, cancer-related death predominated over death due to cardiovascular disease [24,27]. This was probably due in no small part to the relatively successful treatment options for chronic CHD. Encouragingly, the studies presented here showed a clear trend toward more invasive coronary therapy even in cancer patients [3,11,25,26,27,28]. Additionally of positive note was the observation that patients with a history of cancer did not have a relevantly increased risk of bleeding, MACCE, or in-hospital mortality compared with patients without cancer (Figure 2). Similarly, active cancer in the prostate, breast, and leukemia was not per se associated with a high risk of surgery (Figure 2). However, the significantly increased risk of bleeding in active colon cancer and active lung cancer should be noted (Figure 2). In the presence of metastases, the expected risk of intervention was inconsistent: whereas metastatic breast cancer was not expected to have significantly increased PCI complications, this was different for the other cancer types presented here [7,12,23,38].

This suggests that cancer-specific complications such as bleeding and thromboembolism should no longer be stated as a general exclusion criterion for an interventional procedure. Better stent technologies, techniques for precise monitoring of PCI outcomes, and experience with dual antiplatelet therapy also provide hope that more high-risk patients can receive invasive coronary therapy [82,84,85]. However, it would also be desirable if a prospective PCI study could confirm the benefits of the new stent technologies in patients at increased risk of bleeding, such as patients with active colon carcinoma. However, CHD treatment of patients with advanced and complex cancers will remain the subject of individual decisions. It is not only in these particularly difficult cases that joint decision making between the treating cardiologists, oncologists, and primary care physicians is necessary in the best interest of the patient. 

## Figures and Tables

**Figure 1 cancers-14-00434-f001:**
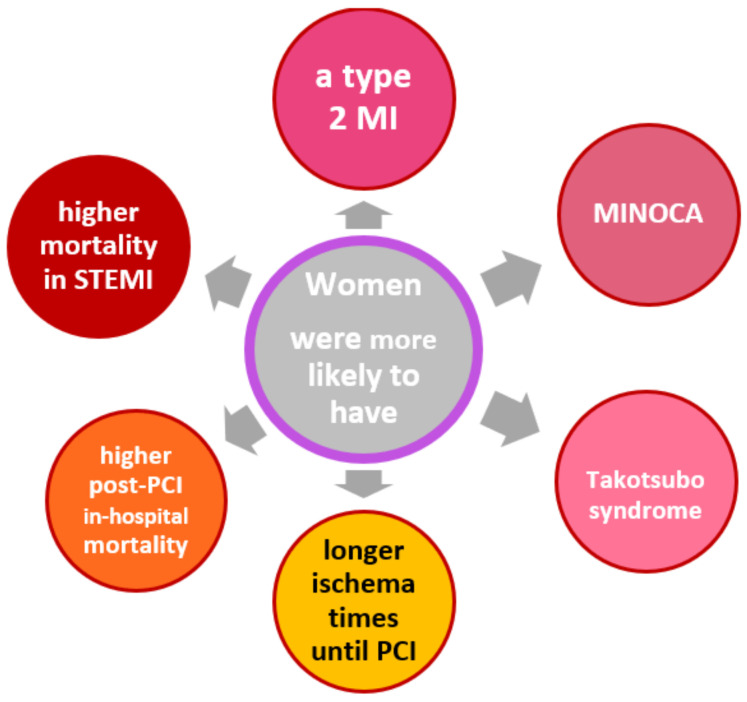
Gender-specific differences in patients with CAD [34]. Abbreviations: MI, myocardial infarction; MINOCA, MI with non-obstructive coronary arteries; PCI, percutaneous coronary intervention; STEMI, ST elevation myocardial infarction.

**Figure 2 cancers-14-00434-f002:**
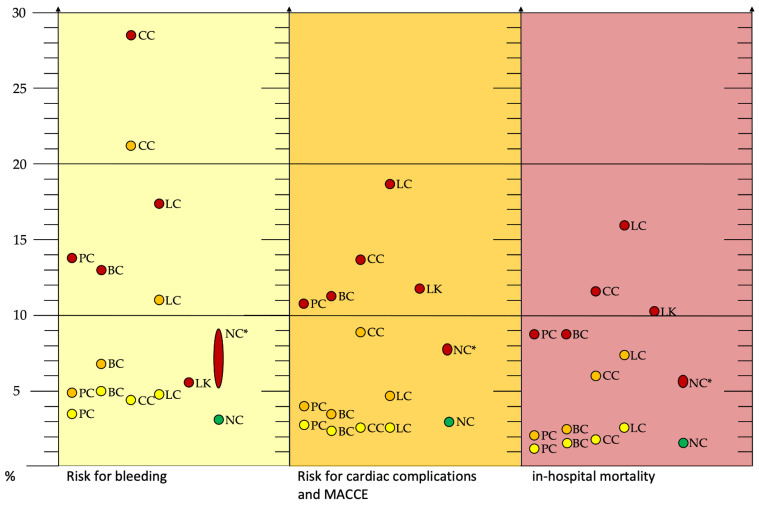
Risk of bleeding (yellow box), cardiac complications and MACCE (orange box), and in-hospital mortality (red box) during PCI and AMI in patients with prostate cancer, breast cancer, colorectal cancer, lung cancer, leukemia, and no cancer. Yellow dots: historical cancer, orange dots: current cancer, red dots: AMI, and green dots: PCI without cancer and AMI. * Non-cancer patients for comparison with source from different studies. Abbreviations: AMI, acute myocardial infarction; BC, breast cancer; CC, colorectal cancer; LC, lung cancer; LK, leukemia; MACCE, major adverse cardiac and cerebrovascular event; PCI, percutaneous coronary intervention; PC, prostate carcinoma; NC, non-cancer [7,14,28].

**Figure 3 cancers-14-00434-f003:**
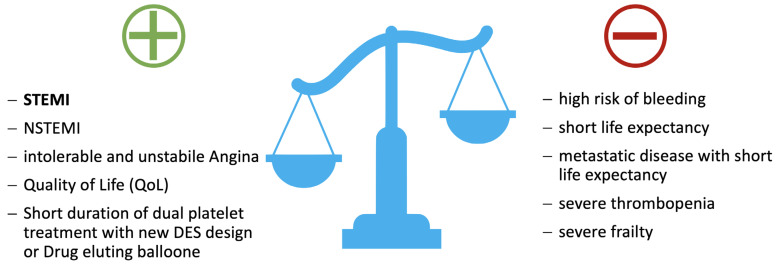
Decision making in favor of or against percutaneous coronary intervention in cancer! Abbreviations: DES, drug eluting stent; NSTEMI, non-ST elevation myocardial infarction; STEMI, ST elevation myocardial infarction.

**Table 1 cancers-14-00434-t001:** Overview of the studies with patients were examined who had coronary artery disease in addition to cancer: The studies in which all patients had cancer are summarized in the upper field; the percentage of CAD and the respective severity of CAD varies depending on the study. The middle section of the table lists the studies in which all patients received coronary intervention. The proportion of patients with simultaneous cancer is correspondingly lower. The studies that cannot be clearly assigned to either the upper or the middle part of the table but were nevertheless analyzed here, are summarized in the lower field of the table.

Author	Year	Study Design	Pat. (n)	ACS/AMI/STEMI	NPL	PCI	Period	Endpoints
Guddati et al. [12]	2016	retrospective registry study, US National Inpatient Database	49,515	100%/67.76%/32.24%	100% (metastatic disease)	STEMI 24.9%; NSTEMI 9.6%	2000–2009	in-hospital mortality, length of hospital stay and discharge disposition
Park et al. [13]	2019	retrospective register study	5300	1.4%/1.075%/0.189%	100%, hematol. Malign.	35.3%	2004–2014	Mortality in-hospital, year 1
Mohamed et al. [14]	2020	registry study	6,750,878	n.i./100%/35.1%	100%, leukaemia	42.9% vs. 28.2% w. leukemia	2004–2014	MACCE and bleeding
Nardi Agmon et al. [15]	2021	single center	3286	60%/-/-	100%	55%/45%	1. 2006–2011 2. 2012–2017	MACE all-cause mortality
			6,808,979					
Kurisu et al. [16]	2012	retrospective, single center	77	AMI 100%	23%	100%	2006–2011	all-cause death year 1
Velders et al. [17]	2013	multicenter, registry study	3423	0%/0%/100%	6.1%	100%	2006–2009	all-cause, cardiac mortality year 1
Wang et al. [18]	2016	retrospective cohort study	2346	n.i./n.i./100%	11.1%	100%	2000–2010	in-hospital and long-term mortality
Landes et al. [19]	2017	retrospective registry study	12,785	n.i.	7.8%	100%	2004–2014	all-cause mortality composite of death, nonfatal mi, target vessel revasc, CABG
Nakatsuma et al. [20]	2018	registry study	12,180	AMI w/o: 36%/C: 29%	9.1%	100%	2005–2007	all-cause death cardiac death, non-cardiac death, heart failure hospitalization, major bleeding, non-CABG surgery, myocardial infarction, definite or probable stent thrombosis, stroke, TLR, any coronary revascularization
Iannaccone et al. [21]	2018	multicenter observational prospective registry; substudy, BleeMACS project	15,401	w/o: 13.2%/28.4%/58.4% C: 16.2%/32.5%/51.3%	6.4%	100%	n.a.	composite event of death and re-infarction y1 bleeding events during follow-up
Potts et al. [7]	2019	registry study	6,571,034	n.i./41.05%/22.84%/ 22.95% (NSTEMI)	1.8% curr., 5.8% prev. C.	100%	2004–2014	Mortality in-hospital in-hospital complication
Gaddam et al. [22]	2020	retrospective cross-sectional study	1,131,415	n.i.	1.27%	100%	2012–2014	risk of association between comorbid cancer and in-hospital mortality in post-PCI inpatients
Kwok et al. [23]	2021	register study	1,933,324	n.i.	9.5%	100%	2010–2014	90-day readmission for AMI 90-day readmission for bleeding
Takeuchi et al. [24]	2021	retrospective, registry study, OASIS	3499	n.i./100%/87.4%	13.2%	100%	1998–2014	Death from cancer, death from cardiac and other causes
			9,685,484					
Pothineni et al. [25]	2017	registry study	3,794,385	n.i./n.i./100%	1.29% (breast 0.15%, lung 0.82%, colon 0.32%)	46.32% (30.8%, 20.2%, 17.3%)	2001–2011	percutaneous coronary intervention (PCI), and in-hospital outcomes in patients
Gong et al. [26]	2018	registry study, observational	270,089	n.i./100%/n.i.	8.48%	w/o vs. w C: 1995, 5.1% vs. 4.3%; 2013, 58.4% vs. 54.4%	1995–2013	Mortality day 30, year1; all-cause mortality overall heart failure, overall myocardial reinfarction, overall stroke
Rohrmann et al. [11]	2018	Multicenter, propensity score matching, AMIS Plus registry	35,249	w/o: 58.1% C: 52.1% STEMI	5.6%	73.4% w/o vs. 67.8% w Cancer	2002-mid 2015	In-hospital outcome
Ederhy et al. [27]	2019	registry study, pospective	3664	n.i./100%/51.34%	6.7%	64.6% w/o, 51.6% w Cancer	2005	5-years mortality
Velders et al. [3]	2020	registry study	175,146	n.i./100%/35.6%	9.3%	48%	2001–2014	All-cause mortality
Bharadwaj et al. [28]	2020	registry study	6,563,255	n.i./100%/36%, 29%	2.8% curr, 6.2% prev C.	43.9% w/o, 21.0% w Cancer	2004–2014	In-hospital mortality and adverse events (MACCE, Bleeding, Stroke)
			10,841,788					

AMI, acute myocardial infarction; C, cancer; CABG, coronary aortic bypass grafting; CAD, coronary artery disease; MACCE, major adverse cardiac and cerebral event; n.i., no information; NSTEMI, Non ST elevation myocardial infarction; prev., previous; STEMI, ST elevation myocardial infarction; w, with; w/o, without angiography.

**Table 2 cancers-14-00434-t002:** Coronary Artery Disease Treatment in Patients with and without Cancer [7,14,28].

Treatment	Prostate Cancer	Breast Cancer	Colon Cancer	Lung Cancer	Leukemia	No Cancer
Coronarangiography (%)	47.5	47.0	44.7	34.8	48.5	64.5–65.2
PCI (%)	29.3	27.4	27.6	21.0	28.2	42.9–43.9
DES (%)	63.3/73.0 *	57.1/73.1 *	38.4/68.9 *	39.3/67.1 *	n.a.	73.7
BMS (%)	31.5/23.0 *	36.1/22.4 *	46.8/26.8 *	49.6/27.6 *	n.a.	21.6
CABG (%)	6.7	4.2	5.1	2.3	6.9	8.9–9.1

Abbreviations: BMS, bare metal stent; CABG, coronary artery bypass grafting; DES, drug eluting stent; PCI, percutaneous coronary intervention. * historical cancer.

**Table 3 cancers-14-00434-t003:** Prevalence of side effects and complications in PCI in Cancer patients [7,14,28].

Treatment	Prostate Cancer	Breast Cancer	Colon Cancer	Lung Cancer	Leukemia ^¢^	No Cancer
PCI in	CC/HC/AMI	CC/HC/AMI	CC/HC/AMI	CC/HC/AMI	AMI	No C/AMI
In-hospital mortality (%)	2.1/1.2/8.7	2.5/1.6/8.7	4.8/1.8/11.6	7.4/2.6/15.9	10.3	1.6/5.7 n.a./5.8 ^¢^
Any complication (%)	11.6/9.2/n.a.	13.7/10.8/n.a.	30.2/10.5/n.a.	19.1/11.0/n.a.	n.a.	8.8/n.a.
Bleeding (%)	4.9/3.5/13.8	6.8/5.0/13.0	21.2/4.5/28.5	11.0/4.8/17.4	5.6	3.1/8.8 n.a./5.3 ^¢^
Vascular complication (%)	0.9/0.8/n.a.	0.7/1.2/n.a.	2.0/0.9/n.a.	1.2/0.8/n.a.	n.a.	1.0/n.a.
Cardiac complications/^#^MACCE (%)	4.0/2.8/10.7 ^#^	3.5/2.4/11.3 ^#^	8.8/2.6/13.7 ^#^	4.7/2.6%18.7 ^#^	0.5/11.8 ^#^	3.0/7.7 ^#^ n.a./0.7/7.8 ^#¢^
Stroke (%)	3.3/3.3/1.9	4.2/3.6/2.4	2.8/3.7/2.1	4.7/3.9/3.5	1.4	2.8/1.7 n.a./1.7 ^¢^

Abbreviations: AMI, acute myocardial infarction; CC, current cancer; HC, historical cancer; MACCE, major adverse cardiac and cerebrovascular event; no C, no cancer; PCI, percutaneous coronary intervention. ^¢^ Leukemia. ^#^ percentage of a MACCE.

**Table 4 cancers-14-00434-t004:** PCI Recommendations in Thrombopenia.

Platelet Count	Additional Considerations	Recommendations
>50,000/mL		− No restrictions
<50,000/mL		− ACT should be monitored during PCI − Prasugrel, Ticagrelor and IIB-IIIA inhibitors should be avoided − Shortening of DAPT duration 2 weeks following POBA alone 4 weeks after BMS 6 months after second or third generation drug-eluting stents (DES) if optimal stent expansion was confirmed by IVUS or OCT
30,000–50,000/mL		− Clopidogrel should be the primary DAPT
<30,000/mL		− revascularization and DAPT should be decided after a preliminary multidisciplinary evaluation (interventional cardiology/oncology/hematology) and a risk/benefit analysis
<20,000/mL	high fever leukocytosis rapid fall in platelet count other coagulation abnormality	Therapeutic platelet transfusions are recommended in thrombocytopenic patients who develop bleeding during or after cardiac catheterization. Repeat platelet counts are recommended after platelet transfusions. 30–50 U/kg unfractionated heparin is the initial recommended dose during PCI.
	In solid tumor patients receiving therapy for − bladder cancer − gynecologic cancer − colorectal cancer − melanoma − necrotic tumors	
<10,000/mL		Aspirin administration should be avoided

Abbreviations: BMS, bare metal stent; DES, drug eluting stent; DAPT, Dual Antiplatelet Therapy; IVUS, intravascular ultrasound; PCI, percutaneous coronary intervention; POBA, percutaneous old balloon angioplasty; OCT, optical coherence tomography modified according to Iliescu CA, et.al. 2016 [81].

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
