# Peer review of "Coronary Artery Disease and Cancer: Treatment and Prognosis Regarding Gender Differences"

_cancers, 2022, doi:10.3390/cancers14020434_

Round 1

Reviewer 1 Report

In this manuscript, Lange & Reineke overview the sex-specific differences in treatment and prognosis of CVD patients with cancer.  While a topic of emerging interest, this review is unfortunately challenging to interpret, based on both a lack of detail for the studies presented, and the rationale for the review itself.  Several suggestions are provided below to enhance clarity and impact for the reader.

  1. While the review generally follows logical structure and sections are relevant to the overarching topic, the sentence structure and word choice throughout the review make interpretation challenging. The authors would benefit from further editing for English grammar and clarity.
    • For example, in the intro, words such as ‘disadvantage’ are unclear and lack specificity. Further, Line 45 is incorrect, ‘even after multivariate regression analysis for age and comorbidities’ should be corrected to simply ‘even after adjustment for age and comorbidities’
  2. The intro also fails to provide a compelling or clear rationale, or even identify which population is being reviewed – cancer patients with CVD? CVD patients?  CVD patients who develop cancer?  A clear line of thought is hard to discern for this reviewer and needs to be addressed.
  3. Critical details have not been included in the review of past studies.  For example, line 66 – ‘The mortality rate and the risk of severe bleeding were higher in AMI patients diagnosed with cancer up to five years ago’.  Relative to who?  Then line 74 – ‘Patients with current cancer diagnosis and AMI had the highest rate of metastatic spreading’ – again, relative to who?  What cancer patients?  What denotes ‘current cancer diagnosis?’ Such details are critical for the reader to interpret these findings.  Such examples are emblematic of issues throughout the MS, which need to be included and logically presented.

Reviewer 2 Report

Lange et al. discuss coronary artery disease in cancer patients taking into account the gender aspect and cancer treatments.

Please find hereby the following comments:

-The authors state that they reviewed studies of over 27 million patients with CAD and cancer. It is unclear whether this manuscript is a systematic review (since it is submitted to the section Systematic Review or Meta-Analysis in Cancer Research), and I miss a description of the search criteria used to select the discussed studies.
-Many abbreviations need to be defined: I highly recommend adding a list of abbreviations. Each abbreviation should be defined in the text.
-The manuscript needs figures and tables to depict and support statements. A graphical abstract is recommended.
-It is not clear what is the intended message from the entire section 2. It simply describes a few studies, but there are no conclusions.
-Section 3: the effect of neoplastic treatments on CVD outcomes has been extensively discussed. This section can be removed and replaced by a few sentences in section 5.
-Section 4: all conclusions need to be supported by mechanistic explanations (maybe from preclinical studies).
-Section 6: the authors mention a “table 4”. However, it is not clear whether the authors are referring to a table in another study or they provide a table.
-Statement in line 498 needs reference(s).
-The manuscript lacks a discussion of a roadmap or suggestions on how to understand the discussed observations (e.g., what studies should be designed in the future?)
-A table summarizing recommendations for the management/treatment of patients for each cancer type/treatment would support the aim of this manuscript.

Reviewer 3 Report

The review by Lange et al. provides an overview of current epidemiological data on the incidence of coronary artery disease in cancer patients.

General remarks:

The review will benefit from revising the beginning of each section. The authors describe in great detail recent epidemiological evidence, but the information is not put into context for a broader readership. The authors should try to include the overall “take-home” message at the beginning of each subsection and provide a brief overall conclusion at the end of each section. Lastly, the review will benefit from a section on coronary artery disease in general and a comparison of corobary artery disease between patients with and without cancer.

Introduction: The authors highlight the gender-specific differences in STEMI and NSTEMI. In line 36 the authors state that “The gender distribution in coronary artery disease is to the disadvantage of men”. However, in the remaining paragraph, the authors correctly describe known gender-specific differences in the diagnosis and incidence of CADs and cancer. The beginning of the paragr              aph should be revised to better reflect the overall content of this section.

Main Text:

  1. Section 3 “Cancer therapy and risk for coronary heart disease” is underdeveloped. The authors should either remove this section or expand to match the length of other sections.
  2. Section 4 “Impact of distinct cancer types”. The authors should provide a justification for the selection of tumors that are discussed in this section. Overall this structure is very helpful and allows comparison of information across different tumors. However, the selection of tumors appears arbitrary without further context.
  3. Section 4: The authors systematically assess each cancer type except for Small intestine cancer. The missing information should be included or section 4.4 should be removed.
  4. Section 4.6: section title is hematologic malignancies, but the authors only discuss leukemia. Later in the paragraph multiple myeloma and (Non-)Hodgkin’s lymphoma appear with some information while the conclusion of this section again only discusses leukemia. This section is not well structured and should be revised accordingly.
  5. Line 319-320 – citation and age ranges are missing for incidence of leukemia in men and women.
  6. Section 5.6: multiple myeloma and  (Non-)Hodgkin’s lymphoma are discussed separately from hematologic malignancies in contrast to section 4. The authors need to homogenize both sections.
  7. Section 5: The authors decided to highlight treatment-related cardiovascular side effects based on different tumors. This section would be more effective if the authors describe each treatment strategy and highlight the different known tumor-specific side effects. The same chemotherapy can be used in the treatment of different tumors. It is intriguing that the risk for coronary artery disease and other cardiovascular side effects is different depending on the tumor and whether the chemotherapy was part of an adjuvant treatment strategy. The authors should provide more insight into the known mechanisms of these therapies, sex-dependent differences, tumors-dependent effects and what is known about the mechanisms for developing CADs.

Round 2

Reviewer 1 Report

The authors should be commended for their rewrite of this manuscript.  Unfortunately, this paper is still very challenging to follow, interpret, and glean any major takeaway messages for this reviewer. 

Major comments:

  1. Many paragraphs remain very challenging to follow. For example, lines 42-55: I believe the purpose is to describe gender differences in CVD outcomes.  This needs to be more structured in its approach – e.g., summary statement at the beginning, describe differences in distribution and risk, then a concluding statement to summarize the salient message (that women with CVD have worse outcomes, and will be the focus of this review?).  This is also applies for other paragraphs (e.g., lines 100-129; lines 130-138, etc…)
  2. Given this paper is not a systematic review or meta-analysis, a line is required explicitly stating what this paper is (e.g. a narrative review?). Based on the last paragraph of the introduction, it is unclear how this data was analyzed and for what exact purpose. This would help lay a foundation for the reader to understand the relevance and rigor applied.
  3. A major challenge in interpreting the data presented in this review is the lack of consistency or grouping of various CV subtypes. Within a given section (for example, section 2), data is presented across numerous CVD subtypes in an inconsistent format, from patients with or without cancer (e.g., gender aspects section). It therefore lacks a consistent thread or theme to aid the reader in interpreting the presented data.
  4. Further to comment 3, as a non-cardiologist, like many readers of this paper, I find the lack of description/acronyms/ lack of structure on CVD subtypes and procedures very hard to interpret, particularly within a cancer context.  Given a large proportion of readers will be coming from the oncology field, it is of critical importance to provide appropriate context and organization to maximize understanding for the reader. 

Minor comments

  1. Line 24 – ‘important’ is an ambiguous word here – most diagnosed? Highest mortality?
  2. Line 35 page 1 requires a reference
  3. Line 41 – ref 5 is incorrect. This should be Ridker 2017 (Lancet). 
  4. Line 56-57- this sentence requires a reference
  5. Line 58-62- this sentence is very hard to follow – a suggestion would be “This review therefore aims to provide clinicians with an overview of how patients, particularly women, with cancer and concomitant coronary artery disease requiring treatment have been treated, We would also like to con- 61 tribute to and how these patients could or should be treated in the future.
  6. Line 84-91 should be deleted as it is repeated in the following paragraph
  7. There does not appear to be a reference to Figure 1

Reviewer 2 Report

The authors have made several edits to the manuscript. However would I like to address the following:

-The manuscript still misses a discussion about the co-occurrence of coronary artery diseases and cancer. Is it due to shared risk factors? Is it due to shared pathophysiology?

-A discussion about surveillance bias is missing.

-The manuscript can still be reduced by at least 20%. Especially section 4:”Specific Cancer Treatment and cardiovascular side-effects.”,  which has been extensively discussed in other reviews.

-Section 5: this section misses a discussion about the direct effect of revascularization procedures on tumor growth. How the management of the Coronary artery disease affect cancer directly? Would it affect cancer progression or tumor growth?

-Figure 1: can be esthetically improved